# Impact of Ready-Meal Consumption during Pregnancy on Birth Outcomes: The Japan Environment and Children’s Study

**DOI:** 10.3390/nu14040895

**Published:** 2022-02-20

**Authors:** Hazuki Tamada, Takeshi Ebara, Taro Matsuki, Sayaka Kato, Hirotaka Sato, Yuki Ito, Shinji Saitoh, Michihiro Kamijima, Mayumi Sugiura-Ogasawara

**Affiliations:** 1Department of Occupational and Environmental Health, Graduate School of Medical Sciences, Nagoya City University, Mizuho-ku, Nagoya 4678601, Japan; h-tamada@med.nagoya-cu.ac.jp (H.T.); ebara@med.nagoya-cu.ac.jp (T.E.); tmatsuki@med.nagoya-cu.ac.jp (T.M.); s-ichiki@med.nagoya-cu.ac.jp (S.K.); h.sato@med.nagoya-cu.ac.jp (H.S.); yukey@med.nagoya-cu.ac.jp (Y.I.); kamijima@med.nagoya-cu.ac.jp (M.K.); 2Department of Pediatrics and Neonatology, Graduate School of Medical Sciences, Nagoya City University, Mizuho-ku, Nagoya 4678601, Japan; ss11@med.nagoya-cu.ac.jp; 3Department of Obstetrics and Gynecology, Graduate School of Medical Sciences, Nagoya City University, Mizuho-ku, Nagoya 4678601, Japan

**Keywords:** stillbirth, ready-made meals, processed foods, birth cohort, pregnancy outcome

## Abstract

Ready-meal consumption is increasing worldwide; however, its impact on human health remains unclear. We aimed to examine the association between processed food and beverage consumption during pregnancy and pregnancy outcomes. Pregnant women were recruited for the Japan Environment and Children’s Study (JECS), a nationwide, large-scale, prospective cohort study. This study included 104,102 registered children (including fetuses or embryos) and collected questionnaire-based data during the first and second/third trimester of pregnancy. Participants’ medical records were transcribed at pregnancy registration, immediately after delivery, and 1 month after delivery. Logistic regression analysis was used to estimate the association between processed food consumption and pregnancy outcomes. The incidence of stillbirth was higher in the group that consumed moderate (1–2 times per week) and high (≥3–7 times per week) amounts of ready-meals (adjusted odds ratio (aOR) = 2.054, 95% confidence interval (CI): 1.442–2.926, *q* = 0.002; aOR = 2.632, 95% CI: 1.507–4.597, *q* = 0.007, respectively) or frozen meals (aOR = 2.225, 95% CI: 1.679–2.949, *q* < 0.001; aOR = 2.170, 95% CI: 1.418–3.322, *q* = 0.005, respectively) than in the group that rarely consumed such foods. Processed food consumption during pregnancy should be carefully considered.

## 1. Introduction

Dietary habits have changed worldwide in the past few decades; the time spent cooking at home has decreased, with more people choosing to frequently dine out [1]. Moreover, the market for ready-made meals has grown, reflecting changes in the social landscape, such as the presence of more women in the workforce and the expanding range of modern life conveniences, which have reduced the time dedicated to domestic work. Concurrently, some studies have suggested that consuming processed foods might be desirable, as it increases the variety of food items consumed [2].

However, the impact of these new dietary choices, such as consuming beverages stored in plastic containers, on human health remains unclear due to inconclusive evidence. Some studies have reported that ready-made meal consumption is associated with an increased risk of overweight and obesity [3]; however, others have reported no such association [4]. The effects of consumption of processed foods during pregnancy on pregnancy outcomes have been largely unexplored; the evidence is confined to one study that suggests that consumption of meals not prepared at home may contribute to the risk of infertility [5]. The prospective birth cohort in Osaka, Japan, showed that the average intake of coffee in pregnant women is 0.14 cups/day [6]. Despite growing interest in the effects of processed food consumption on human health, few studies have examined the association between the intake of such meals and human health outcomes and specifically birth outcomes, such as stillbirth and pre-term birth, as well as on developmental measures, such as size (small for gestational age (SGA)) and low birth weight.

Elucidating the impact of processed food consumption on pregnancy-related out-comes is relevant to the formulation of public health policies, as evidence suggests that consumption of such foods is associated with exposure to chemicals, such as bisphenol A (BPA) and Di (2-ethylhexyl) phthalate (DEHP), that may disrupt endocrine function. Our previous study showed that serum BPA levels were significantly higher in women who had recurrent miscarriages than in controls [7]. Meanwhile, in a separate study, we found no evidence of an association between recurrent miscarriages and exposure to polychlorinated biphenyls, hexachlorobenzene, or 1,1,1-trichloro-2,2-bis (p-chlorophenyl) ethylene metabolite of 1,1-dichloro-2,2-bis (p-chlorophenyl) ethylene [8]. The present study aimed to examine the association between processed foods, including beverages stored in a can or plastic containers; their consumption during pregnancy; and pregnancy outcomes, using data from a large Japanese birth cohort study.

## 2. Materials and Methods

Pregnant women were recruited between January 2011 and March 2014 for the Japan Environment and Children’s Study (JECS), a nationwide, large-scale, prospective cohort study registered in the UMIN Clinical Trials Registry (number UMIN000030786). Expectant mothers were eligible for the present study if they resided in the study area at the time of recruitment, had their due date after 1 August 2011, and were fluent Japanese speakers who could understand and complete a set of self-administered questionnaires [9,10,11]. The sample size was calculated prospectively by the JECS Working Group and presented in the JECS protocols ahead of recruitment. Participants’ medical records were transcribed by physicians or Research Coordinators at registration, immediately after delivery, and at 1 month after delivery.

This study was based on the jecs-ag-20160424 dataset, which included 104,102 registered children (fetuses and embryos), and was released to all researchers involved with the JECS in June 2016. In cases of multiple pregnancies, outcomes of the second and third pregnancies were excluded to remove duplicates (*n* = 1003 (0.96%)). Infants born with physical anomalies were excluded (*n* = 9008 (8.7%)). Twenty-nine (0.03%) participants withdrew from the study. Finally, data from 94,062 pregnancies were included in the main analysis (Figure 1). The mean (SD) age at registration was 30.7 (5.1) years. The mean (SD) gestational age at registration was 14.4 (5.6) weeks.

### 2.1. Ethical Approval

The JECS protocol was reviewed and approved by the Ministry of the Environment Institutional Review Board on Epidemiological Studies and by the Ethics Committees of all participating institutions. Written informed consent was obtained from all participants. The study was conducted in accordance with the Helsinki Declaration and other national regulations and guidelines.

### 2.2. Variables

Study participants completed questionnaires in the first and second/third trimesters of pregnancy, providing data on socio-demographic, socioeconomic, and life-style characteristics. Data transcribed from medical records at the time of study enrollment included information on maternal age, body weight, and height; use of in vitro fertilization and embryo transfer (IVF-ET) for the present pregnancy; and obstetric history. Data transcribed from medical records immediately after delivery included details of maternal and gestational age, single/multiple pregnancy status, and pregnancy outcomes, such as live/stillbirth, miscarriage/induced abortion, and the mode of delivery (vaginal delivery vs. cesarean section), pregnancy-related complications, perinatal outcomes, and infant sex and weight. Data transcribed from medical records 1 month after delivery included maternal age and information on birth defects.

### 2.3. Outcomes

The outcomes of interest were stillbirth, pre-term birth, SGA, and low birth weight. In this study, pre-term birth was defined as delivery between 22 and <37 weeks of gestation. SGA was defined as a birth weight below the 10th percentile, according to the new Japanese neonatal anthropometric charts for gestational age at birth.

### 2.4. Exposures and Covariates

The frequency of eating ready-made meals (pre-packed foods sold at convenience stores, supermarkets, or boxed lunch shops), frozen foods, retort pouch foods, convenience foods (instant noodles, soups, or other foods packed in plastic cups that can be cooked by pouring hot water), and canned foods during second/third trimesters was categorized as <1 time per week, 1–2 times per week, and 3–7 or more times per week based on self-reported lifestyle data [12]. In general, ready-made and frozen meals were defined as foods that require microwave preparation, while retort pouch and convenience foods were those that require cooking in boiling water either with or without packaging (e.g., curry or cup noodles, respectively).

In addition, we assessed the frequency of coffee consumption, including the type of container when purchased (e.g., a can or plastic bottle) and the mode of preparation by the consumer. The frequency of black, green, and oolong tea consumption was also evaluated, including whether it was in a can or plastic bottle and if the tea was made from tea leaves by the consumer. To estimate odds ratios (OR), beverage consumption was recalculated separately depending on the source of beverage (can or bottle vs. beans or leaves). Frequency categories were recorded as follows: “less than once a week” became “once a week”, “once or twice a week” became “twice a week”, “three to four times per week” became “four times per week”, “five to six times per week” became “six times per week”, “a cup daily” became “seven times per week”, “2–3 cups daily” became “21 times per week”, “4–6 cups daily” became “42 times per week”, “7–9 cups daily” became “63 times per week”, and “>10 cups daily” became “70 times per week”. Based on self-reported data, the total beverage consumption was defined as <7 times/week, 7–13 times/week, and ≥14 times/week.

The covariates of interest included maternal age at registration, body mass index, IVF-ET status, maternal smoking and drinking status, income level, maternal educational status, history of pregnancy loss, parity, maternal working hours, hypertensive disorders of pregnancy, gestational diabetes, maternal energy intake calculated using the Food Frequency Questionnaire [13] for pregnant women from the JECS data, and the consumption of processed foods and beverages that were beyond the focus of this analysis.

### 2.5. Statistical Analysis

Descriptive statistics were reported as frequencies. Logistic regression analysis was used to estimate the association between dietary habits and each pregnancy outcome. The OR of each pregnancy outcome was adjusted for covariates. Crude and adjusted odds ratios (aOR) or mean differences with 95% confidence intervals (95% CI) were reported, as suitable. Missing values were handled with multiple imputation methods. To correct the false discovery rate, the *q*-value was obtained using the Benjamini–Hochberg procedure in R statistical software (version 3.5.2). A *q*-value of <0.05 was considered statistically significant. Multiple correspondence analysis (MCA) and hierarchical cluster analysis (HCA; Ward’s method) were used to investigate the association between dietary habits and beverage intake. All analyses were conducted using SPSS version 23 (IBM Corp., Japan) except for *q*-value estimates.

## 3. Results

The participants’ maternal characteristics and pregnancy outcomes are presented in Table 1. Most participants reported consuming processed foods, such as ready-made meals (58.8%), frozen meals (62.9%), retort pouch (72.9%), convenience (74.8%), and canned (87.0%) foods, at a frequency of <1 time per week. In addition, most participants declared that they rarely (<7 times per week) drank beverages from a can or plastic bottle (60.0%) or those extracted from coffee beans or tea leaves (52.5%).

There was a significant association between the incidence of stillbirth and the consumption of processed foods in all categories except canned food (Appendix A). The incidence of stillbirth increased in the group reporting moderate (1–2 times per week) consumption of ready-made (OR = 3.217, 95% CI: 2.371–4.364, *q* < 0·001) and frozen (OR = 3.404, 95% CI: 2.633–4.400, *q* < 0.001) meals and retort pouch (OR = 2.266, 95% CI: 1.801–2.851, *q* < 0.001) and convenience (OR = 2.369, 95% CI = 1.783–3.147, *q* < 0.001) foods. After adjusting for covariates, this association remained significant for ready-made and frozen meals but not for retort pouch or convenience foods (Table 2). The incidence of stillbirth was higher in the group reporting moderate (1–2 times per week) and high (≥ 3–7 times per week) consumption of ready-made (aOR = 2.054, 95% CI: 1.442–2.926, *q* = 0.002; aOR = 2.632, 95% CI: 1.507–4.597, *q* = 0.007, respectively) and moderate and high consumption of frozen (aOR = 2.225, 95% CI: 1.679–2.949, *q* < 0.001; aOR = 2.170, 95% CI: 1.418–3.322, *q* = 0.005, respectively) meals than in the group that consumed such foods less often than once per week. In addition, the incidence of pre-term birth was higher in the group reporting moderate (1–2 times per week) consumption of ready-made meals (aOR = 1.100, 95% CI: 1.024–1.181, *q* = 0.030), and the incidence of low birth weight was higher in the group reporting moderate consumption of retort pouch foods (aOR = 1.105, 95% CI: 1.035–1.180, *q* = 0.012) than that in the low consumption (less than once a week) group.

Beverage consumption was associated with pregnancy outcomes such as stillbirth, pre-term birth, SGA infant, and low birth weight. In particular, the incidence of stillbirth was higher in the group reporting moderate (7–13 times per week) and high (≥ 14 times per week) consumption of beverages (can or plastic bottle) (aOR = 3.484, 95% CI: 2.611–4.649, *q* < 0.001; aOR = 2.930, 95% CI: 1.837–4.673, *q* < 0.001, respectively) than that in the group reporting low consumption. In addition, the incidence of stillbirth increased in the group reporting moderate and high consumption of beverages extracted from beans or leaves (aOR = 3.752, 95% CI: 2.923–4.816, *q* < 0.001; aOR = 1.754, 95% CI: 1.192–2.581, *q* = 0.021, respectively). A similar trend was observed in the incidence of pre-term birth. The incidence of SGA infants increased in drinkers of coffee or tea from bean or leaves, and that of low-birth weight increased in drinkers of coffee or tea from cans or plastic bottles relative to the estimates of the low-consumption group.

Estimates could not be obtained for the high-consumption group due to the small sample size.

We examined dietary habits in MCA after dividing it into four clusters by HCA (Figure 2). Cluster A included high consumption of retort pouch, convenience, and canned foods. Cluster B included high consumption of ready-made and frozen meals. Cluster C included moderate consumption of ready-made, frozen meals, retort pouch, convenience, and canned foods. Cluster D included low consumption of all kinds of processed foods and all frequency groups of both beverages (in a can or plastic bottle and extracted from coffee bean or tea leaf). Clusters B and C represented the dietary pattern associated with a high risk of stillbirth, specifically moderate to high consumption of ready-made and frozen meals. The increase in the risk of stillbirth rates associated with consumption of ready-made and frozen meals was independent of the impact of beverage consumption in both cluster and covariate-adjusted analyses (Table 2).

## 4. Discussion

To the best of our knowledge, this is the first large-scale birth cohort study to show that consumption of ready-made and frozen meals during pregnancy might increase the risk of stillbirth. Four kinds of processed foods were associated with an increased risk of stillbirth in the crude analysis (Appendix A); however, consumption of foods that require microwave heating was significantly associated with a risk of stillbirth after adjusting for covariates (Table 2).

These findings suggest that food packaging and reheating methods may affect outcomes possibly through exposure to chemicals present in meal packaging that are released in the process of cooking in a microwave [14]. BPA, which is used in food packaging, may be a chemical particularly associated with the risk of stillbirth. BPA has a tolerable intake of 0.05 mg/kg b.w./day, as defined by the European Food Safety Authority [15]. However, microwave cooking has been reported to augment BPA migration [16]. The annual intake of BPA from canned foods among the Japanese population has been estimated as 644 ng/person/day in 2011–2012 [17]. BPA can be used as an antioxidant or as a plasticizer in the production of polypropylene, polyethylene, poly-vinyl chloride, and polycarbonate, which are often used in food packaging [18].

Previous studies have shown that BPA exposure may affect human reproductive health even at doses lower than the tolerable daily intake [19]. Several studies have replicated our previous findings on the association between serum BPA levels and the risk of recurrent miscarriage [7,20]. Moreover, Allard et al. showed that exposure to BPA increased the rate of sterility and embryonic death in a mammalian model, where it impaired chromosome synapsis and disrupted the meiotic double-strand break-repair progression [21]. In addition, our previous study showed evidence of a dose-response association between BPA concentration and miscarriage risk mediated by embryonic aneuploidy or antinuclear antibody positivity [7]. Lathi et al. suggested that BPA concentration values may be used to predict the risk of both embryonic euploid and aneuploid miscarriage [20].

In the present study, an association with stillbirths at ≥12 weeks’ gestation was found. It remains unclear how consumption of ready-made and frozen meals increases the risk of stillbirths and whether BPA exposure is involved; the risk of aneuploidy decreases with the increase in gestational age. Furthermore, ready-made meal consumption may increase exposure to phthalates, including DEHP [22], which has been weakly associated with an increased risk of adverse outcomes, such as miscarriage and pre-term birth [23]. Studies using animal models have recently shown the combined adverse effects of BPA and DEHP on gestational outcomes [24]. Styrene oligomers may also be released from food packaging; however, this substance has not been associated with human reproductive toxicity to date [25].

This study revealed an association between beverage intake and the risk of stillbirth. Beverage intake was categorized as out of “a can or plastic bottle” and “extracted from coffee bean or tea leaf”; these categories were associated with ORs of 3.548 and 3.703, respectively. In addition, the consumption of these beverage types was associated with an increased risk of pre-term birth, SGA, and low birth weight. In general, canned and bottled beverages contain large amounts of sugar; consequently, our analyses were adjusted for energy consumption as a surrogate measure of sugar intake. Moreover, caffeine intake has been associated with pregnancy loss, pre-term birth, and low birth weight [26]; therefore, the World Health Organization recommends that daily coffee consumption not exceed 3–4 cups during pregnancy. A meta-analysis has shown that heavy caffeine intake increased the risk of pregnancy loss, including stillbirth [27]. The present findings are consistent with those of previous studies.

The present study has three primary strengths. First, it included a large sample of approximately 100,000 participants. Second, the participants were representative of the Japanese pregnant population since the JECS covered rural and urban areas across Japan [11]. Third, this study is potentially relevant to social policy; socioeconomic and environmental factors associated with processed food consumption should be considered alongside their health effects. It has been suggested that socioeconomic status is related to processed food consumption and resultant increased obesity rates; poor health literacy has been implicated as the source of unhealthy lifestyle choices, including poor diet [28]. Environmentally, the increase in processed food consumption has raised concerns [29]; in response, the concept of “ethical consumption” has been advocated, promoting sustainable consumption behavior. As a result, reducing processed food consumption falls within the scope of the Sustainable Development Goals [30]. Overall, the present study suggests that the development of a processed food and beverage consumption policy requires a comprehensive and multidisciplinary approach, as it has consequences for several fields outside of human health; the present study is one such contribution.

The main limitation of this study is that it did not identify underlying mechanisms or substances directly associated with the reported increases in risk; in addition, we did not measure biochemical parameters, such as BPA levels, which should be quantified in future studies. In addition, poor quality, such as excess sugar, fat, saturated fat intake, and lack of dietary fiber, is a problem in a processed-food-rich diet [31], and there is a possibility of residual confounding regarding the effect of this on outcomes. However, since energy intake is adjusted as confounding in this study, it is considered that the effect of food components that correlate with energy intake is adjusted. Finally, the incidence of early miscarriage could not be examined, as the mean gestational age at registration was 14.4 weeks.

## 5. Conclusions

Our findings suggest that processed food and beverage consumption during pregnancy increases the risk of adverse pregnancy outcomes, including stillbirth. This finding may result from exposure to chemicals contained in food packaging, which may increase with microwave cooking. The present findings suggest that dietary and nutritional advice should be included in prenatal counseling to help prevent serious adverse outcomes [32]. The environmental impact of processed food consumption should be examined, and future studies should examine biochemical parameters, including urine samples, and their impact on the risk of adverse pregnancy outcomes.

## Figures and Tables

**Figure 1 nutrients-14-00895-f001:**
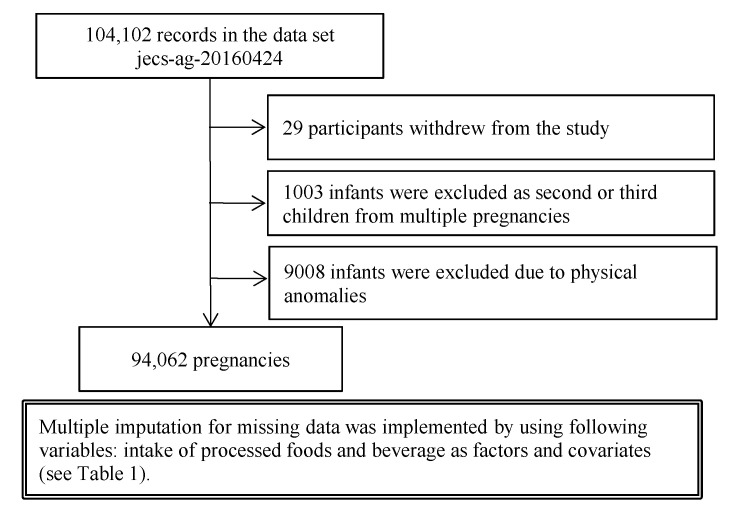
Flow diagram summarizing the study recruitment process.

**Figure 2 nutrients-14-00895-f002:**
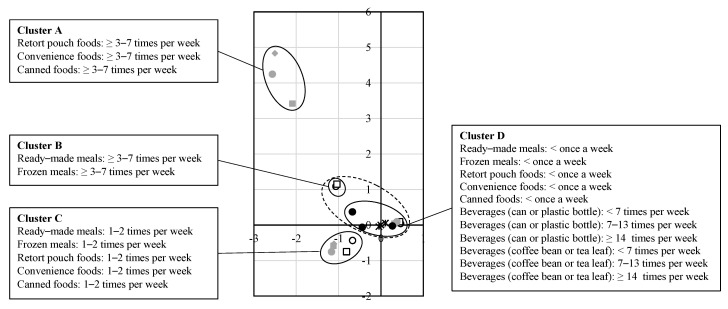
The results of multiple correspondence analysis.

**Table 1 nutrients-14-00895-t001:** Maternal characteristics of participants and incidence of events (*N* = 94,062).

Variables		*n*	(%)
Stillbirth (≥12 weeks gestation)		842	(0.9)
Pre-term birth (<37 weeks gestation)		4547	(4.8)
Small for gestational age infant		6599	(7.0)
Low birth weight (<2500 g)		7601	(8.1)
Maternal age at registration			
	<20	1131	(1.2)
	20—29	37,882	(40.3)
	30–39	51,554	(54.8)
	≥40	3263	(3.5)
	Missing	232	(0.2)
Smoking histories during second/third trimester			
	Non-smokers	51,049	(54.3)
	Ex-smokers who quit before pregnancy	21,183	(22.5)
	Ex-smokers who quit during early pregnancy	12,186	(13.0)
	Current smokers	4049	(4.3)
	Missing	5595	(5.9)
Maternal educational status			
	Junior high school or high school	32,362	(34.4)
	Higher professional school or professional school	37,256	(39.6)
	Junior college or college	17,789	(18.9)
	Postgraduate college	1285	(1.4)
	Missing	5370	(5.7)
Annual income (JPY × 10,000)			
	<200	4746	(5.0)
	200–<400	28,775	(30.6)
	400–<600	27,330	(29.1)
	600–<800	13,080	(13.9)
	800–<1000	5362	(5.7)
	≥1000	3489	(3.7)
	Missing	11,280	(12.0)
Alcohol intake during second/third trimesters			
	Never	29,632	(31.5)
	Abstinence before pregnancy	15,196	(16.2)
	Abstinence from this pregnancy	41,171	(43.8)
	Continuance drinking	2477	(2.6)
	Missing	5586	(5.9)
In vitro fertilization and embryo transfer			
	No	90,591	(96.3)
	Yes	2887	(3.1)
	Missing	584	(0.6)
Maternal BMI			
	<18.5	15,080	(16.0)
	18.5–<25.0	68,255	(72.6)
	≥ 25.0	10,050	(10.7)
	Missing	677	(0.7)
Histories of pregnancy loss			
	Never	71,555	(76.1)
	Once	16,297	(17.3)
	Twice	3604	(3.8)
	More than 3 times	1055	(1.1)
	Missing	1551	(1.6)
Histories of live birth			
	No	36,792	(39.1)
	Yes	54,424	(57.9)
	Missing	2846	(3.0)
Hypertensive disorders of pregnancy			
	No	87,754	(93.3)
	Yes	2759	(2.9)
	Missing	3549	(3.8)
Gestational diabetes			
	No	89,570	(95.2)
	Yes	943	(1.0)
	Missing	90,513	(96.2)
Frequency of ready-made meals			
	<once a week	55,354	(58.8)
	1–2 times per week	25,342	(26.9)
	≥3–7 times per week	8030	(8.5)
	Missing	5336	(5.7)
Frequency of frozen meals			
	<once a week	59,163	(62.9)
	1–2 times per week	20,747	(22.1)
	≥3–7 times per week	8709	(9.3)
	Missing	5443	(5.8)
Frequency of retort pouch foods			
	<once a week	68,609	(72.9)
	1–2 times per week	18,200	(19.3)
	≥3–7 times per week	1712	(1.8)
	Missing	5541	(5.9)
Frequency of convenience foods in plastics container			
	<once a week	70,390	(74.8)
	1–2 times per week	16,196	(17.2)
	≥ 3–7 times per week	1934	(2.1)
	Missing	5542	(5.9)
Frequency of canned foods			
	<once a week	81,808	(87.0)
	1–2 times per week	6070	(6.5)
	≥3–7 times per week	307	(0.3)
	Missing	5877	(6.2)
Frequency of beverages (can or plastic bottle)			
	<7 times per week	56,446	(60.0)
	7–13 times per week	22,676	(24.1)
	≥14 times per week	7284	(7.7)
	Missing	7656	(8.1)
Frequency of beverages (coffee bean or tea leaf)			
	<7 times per week	49,422	(52.5)
	7–13 times per week	21,666	(23.0)
	≥14 times per week	15,179	(16.1)
	Missing	7795	(8.3)
Maternal working hour (h), mean (SD)		4.0	(4.0)
Maternal energy intake (kcal/day), mean (SD)		1715.2	(647.4)

JPY, Japanese yen; BMI, body mass index; SD, standard deviation.

**Table 2 nutrients-14-00895-t002:** Maternal characteristics of participants and incidence of events (*N* = 94,062).

		Stillbirth (≥12 Weeks Gestation) ^a^		Pre-Term Birth (<37 Weeks Gestation) ^b^		Small for Gestational Age Infant		Low Birth Weight (<2500 g)	
		*n* = 842		*n* = 4547		*n* = 6599		*n* = 7601	
		*q*-value	Adjusted ORs (95% CI)	*q*-value	Adjusted ORs (95% CI)	*q*-value	Adjusted ORs (95% CI)	*q*-value	Adjusted ORs (95% CI)
**Ready-made meals**	**<once a week**		1.000		1.000		1.000		1.000
**(cooked with microwave heating in general)**	**1–2 times per week**	0.002	2.054 (1.442–2.926)	0.030	1.100 (1.024–1.181)	0.050	0.929 (0.874–0.987)	0.840	0.990 (0.936–1.048)
	**≥3–7 times per week**	0.007	2.632 (1.507–4.597)	0.950	0.993 (0.877–1.125)	0.375	0.940 (0.853–1.036)	0.542	0.961 (0.875–1.056)
**Frozen meals**	**<once a week**		1.000		1.000		1.000		1.000
**(cooked with microwave heating in general)**	**1–2 times per week**	0.000	2.225 (1.679–2.949)	0.231	1.068 (0.985–1.158)	0.542	1.026 (0.962–1.095)	0.449	1.034 (0.971–1.102)
	**≥3–7 times per week**	0.005	2.170 (1.418–3.322)	0.099	1.126 (1.005–1.261)	0.781	1.020 (0.93–1.119)	0.961	1.003 (0.918–1.097)
**Retort pouch foods**	**<once a week**		1.000		1.000		1.000		1.000
**(heated with boiling water in general)**	**1–2 times per week**	0.542	1.123 (0.856–1.475)	0.498	1.043 (0.957–1.135)	0.283	1.052 (0.982–1.128)	0.012	1.105 (1.035–1.18)
	**≥3–7 times per week**	0.077	0.312 (0.109–0.891)	0.161	0.786 (0.605–1.021)	0.542	1.081 (0.89–1.313)	0.242	1.155 (0.963–1.385)
**Convenience foods in plastic container**	**<once a week**		1.000		1.000		1.000		1.000
**(heated with boiling water in general)**	**1–2 times per week**	0.296	1.265 (0.908–1.762)	0.388	1.052 (0.966–1.144)	0.542	1.031 (0.96–1.108)	0.375	1.043 (0.975–1.115)
	**≥3–7 times per week**	0.050	0.391 (0.18–0.849)	0.449	0.886 (0.707–1.11)	0.542	0.927 (0.769–1.117)	0.619	0.943 (0.793–1.121)
**Canned foods**	**<once a week**		1.000		1.000		1.000		1.000
**(without heating in general)**	**1–2 times per week**	0.911	1.047 (0.666–1.646)	0.050	1.157 (1.027–1.305)	0.936	1.008(0.908–1.119)	0.231	1.082 (0.983–1.191)
	**≥3–7 times per week**	NA	NA NA	0.388	0.694 (0.378–1.277)	0.911	0.956 (0.61–1.499)	0.888	0.946 (0.623–1.435)
**Beverage (can or plastic bottle)**	**<7 times per week**		1.000		1.000		1.000		1.000
	**7–13 times per week**	0.000	3.484 (2.611–4.649)	0.012	1.125 (1.042–1.214)	0.652	0.981 (0.922–1.044)	0.027	1.084 (1.022–1.149)
	**≥14 times per week**	0.000	2.930 (1.837–4.673)	0.000	1.294 (1.160–1.444)	0.161	1.091 (0.993.–1.199)	0.006	1.160 (1.062–1.268)
**Beverage (coffee bean or tea leaf)**	**<7 times per week**		1.000		1.000		1.000		1.000
	**7–13 times per week**	0.000	3.752 (2.923–4.816)	0.053	1.094 (1.014–1.180)	0.007	1.105 (1.040–1.175)	0.027	1.082 (1.022–1.147)
	**≥14 times per week**	0.021	1.754 (1.192–2.581)	0.542	0.965 (0.884–1.054)	0.001	1.150 (1.073–1.232)	0.375	1.043 (0.975–1.115)

Adjusted for maternal age at registration, smoking histories of second/third trimester, maternal educational status, annual income, alcohol intake of second/third trimester, in vitro fertilization and embryo transfer, maternal BMI, histories of pregnancy loss, parity, working hours, energy intake, hypertensive disorders of pregnancy, gestational diabetes, and each dietary habit. The odds ratio was calculated with the data complemented by the multiple imputation method. ^a^ Early miscarriage (<12 weeks’ gestation) and artificial abortion were excluded from analyses. ^b^ Miscarriage and artificial abortion were excluded from analyses.

## Data Availability

Data are unsuitable for public deposition due to ethical restrictions and legal framework of Japan. It is prohibited by the Act on the Protection of Personal Information (Act No. 57 of 30 May 2003, amendment on 9 September 2015) to publicly deposit data containing personal information. Ethical Guidelines for Medical and Health Research Involving Human Subjects enforced by the Japan Ministry of Education, Culture, Sports, Science, and Technology and the Ministry of Health, Labour, and Welfare also restrict the open sharing of epidemiologic data. All inquiries about access to data should be sent to jecs-en@nies.go.jp. The person responsible for handling enquiries sent to this e-mail address is Dr. Shoji F. Nakayama, JECS Program Office, National Institute for Environmental Studies.

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
