# Peer review of "Impact of Ready-Meal Consumption during Pregnancy on Birth Outcomes: The Japan Environment and Children’s Study"

_nutrients, 2022, doi:10.3390/nu14040895_

Round 1

Reviewer 1 Report

Thanks for the opportunity to review this important article. Please find attached comments for your consideration. 

Impact of ready-meal consumption during pregnancy on birth 2 outcomes: The Japan Environment and Children's Study
An exciting and informative study.
Please check these two statements in the results:
... The incidence of stillbirth increased in the group reporting moderate (1–2 times per week) consumption of ready-made and frozen meals (line 159 – 160)
AND
...the incidence of stillbirth was higher in the group reporting moderate (7–13 times per week) and high (≥ 14 times per week) consumption of beverages (can or plastic bottle) (line 177 – 179)

Is there a possibility of collinearity of your variables? For example, could it be the case that it is EITHER ready-made/frozen meals OR beverages rather than each variable having an effect on its own? Has your research established this? If not, you may consider adding this to the study limitations.

Please check grammar/spellings

Line 47   ...such as
Line 80    text box below the figure
Line 278  full stop in red font.
Line 189  Table 2. Please remove rows for <once a week and < 7 times per week. The data is zero throughout and does not inform the results.
Line 8     To make the discussion easier to read, you could start new paragraphs at lines 215, 224, and 234.

Author Response

Response Letter to the Reviewer 1

I greatly appreciated the critical review and have revised the manuscript in response to the reviewers’ suggestions as detailed below.

Please check these two statements in the results:
... The incidence of stillbirth increased in the group reporting moderate (1–2 times per week) consumption of ready-made and frozen meals (line 159 – 160)
AND
...the incidence of stillbirth was higher in the group reporting moderate (7–13 times per week) and high (≥ 14 times per week) consumption of beverages (can or plastic bottle) (line 177 – 179)

Is there a possibility of collinearity of your variables? For example, could it be the case that it is EITHER ready-made/frozen meals OR beverages rather than each variable having an effect on its own? Has your research established this? If not, you may consider adding this to the study limitations.

<Response>  

We appreciate you taking the time to offer us your comments and insights related to the paper.
As you pointed out, considering the mutual effect of collinearity of each variable is important. In this regard, we confirmed variance inflation factor (VIF), one of the metrics for collinearity, prior to the logistic regression analyses. the VIFs were sufficiently low, so we've already solved it. Thank you.

Please check grammar/spellings

Line 47   ...such as

<Response> We corrected it, thank you.
Line 80    text box below the figure

<Response> Thank you for your careful confirmation. We revised it.
Line 278  full stop in red font.

<Response> We made it black, thank you.
Line 189  Table 2. Please remove rows for <once a week and < 7 times per week. The data is zero throughout and does not inform the results.

<Response> Thank you for your informative suggestion. Since “<once a week” and “< 7 times per week” are reference categories in the analysis, we inserted “1.000” as references, instead of deleting these rows.

Line 8     To make the discussion easier to read, you could start new paragraphs at lines 215, 224, and 234.

<Response> We added line breaks for them.

Reviewer 2 Report

This Japanese observational study evaluates the impact of processed food and beverage intake and pregnancy outcomes. The results of this study indicate that moderate and high intake of ready-meals increases the risk of stillbirth and that beverage consumption is associated with stillbirth, SGA, low birth weight and preterm birth. I have the following comments:

  1. In the introduction section the authors should indicate the incidence of stillbirth, SGA, preterm birth and low birth weight in the general Japanese population
  2. The recruitment of participants finished almost 8 years ago (March 2014). Why did the authors not perform the analysis earlier? Is any long-term follow up information of study participant available?
  3. What is the average intake of coffee in pregnant women population in Japan?
  4. What nutritional advice do pregnant women in Japan get in the early pregnancy?

Author Response

Response Letter to the Reviewer 2

Thank you for the points raised that have helped us improve the quality of our manuscript. Please find below our detailed responses.

  1. In the introduction section the authors should indicate the incidence of stillbirth, SGA, preterm birth and low birth weight in the general Japanese population

<Response>  

We appreciate your important comments.

There are several manuscripts concerning the incidence of preterm birth in the Japanese population. 
It was 5.4 % (excluding stillbirths) and 5.5 % (including stillbirths) in Morizaki’s study. However, the sample size (3496) was relatively small. Takeuchi et al. indicated 5.4% (2,005/36,885) in 2010.

Thus, we calculated using the open database provided by the Ministry of Health, Labor and Welfare, Japan as below.

Vital Statistics | File | Browse Statistics | Portal Site of Official Statistics of Japan (e-stat.go.jp)

There were 1,050,807 live-born infants (including multiple pregnancies) and 11,940 stillbirths (12 weeks’ gestation) in 2011. The number of preterm births and low birth weight were 60,285 and 100,378 (including multiple pregnancies). The number of mothers was 1,066,130. The incidences of stillbirth, preterm birth and low birth weight were nearly 1.12%, 5.67% and 9.45% in Japanese population. The incidences in the present study were 0.9%, 4.8% and 8.1%, respectively. The main reason for relatively higher incidences in the general Japanese population is that stillbirths, preterm births, and low birth weights included multiple pregnancies. Furthermore, the mean gestational age at registration was 14.4 wks (SD:5.6), yielding bias to underestimation of stillbirth. If such data were presented in the Introduction, it might mislead the reader. This study did not focus on the incidence perse, rather the association between processed food consumption during pregnancy and pregnancy outcomes, using the data without early stillbirth, physical abnormalities. As such, we have decided to refrain from introducing their incidence in the introduction. We hope for your kind understanding.

Morisaki N, Ganchimeg T, Vogel JP, Zeitlin J, Cecatti JG, Souza JP, Pileggi Castro C, Torloni MR, Ota E, Mori R, Dolan SM, Tough S, Mittal S, Bataglia V, Yadamsuren B, Kramer MS; PREBIC Epidemiology Working Group and the WHO-MCS Research Network. Impact of stillbirths on international comparisons of preterm birth rates: a secondary analysis of the WHO multi-country survey of Maternal and Newborn Health. BJOG. 2017 Aug;124(9):1346-1354. doi: 10.1111/1471-0528.14548. Epub 2017 Feb 20.PMID: 28220656 

Takeuchi A, Namba T, Matsumoto N, Tamai K, Nakamura K, Nakamura M, Kageyama M, Kubo T, Tsukahara H, Yorifuji T. Preterm birth and Kawasaki disease: a nationwide Japanese population-based study. Pediatr Res. 2021 Oct 8. doi: 10.1038/s41390-021-01780-4. Online ahead of print.PMID: 34625654

  1. The recruitment of participants finished almost 8 years ago (March 2014). Why did the authors not perform the analysis earlier? Is any long-term follow up information of study participant available?

<Response>

Since this study is a part of multiinstitutional joint research, we have some regulations to treat the data. We needed to start the analysis after the data set was cleaned, that is, in large epidemiological research, data cleaning is required to identify and correct errors or at least to minimize their impact on study results. After that, the data were distributed and available to the research members. This is one reason why there was such a time lag.
As for the long-term follow-up information, we have no further information on pregnant women (participants of this study) because the JECS birth cohort mainly focuses on children's development.

  1. What is the average intake of coffee in pregnant women population in Japan?

<Response>

We added the following sentence to the introduction.
The prospective birth cohort in Osaka, Japan showed that the average intake of coffee in pregnant women is 0.14 cups/day.

  1. What nutritional advice do pregnant women in Japan get in the early pregnancy?

<Response>

Thank you for your question. In Japan, the Ministry of Health, Labor and Welfare has formulated "Dietary Guidelines for Pregnant Women," which advices on the nutrients that should be consumed during pregnancy. Guideline by the Japan Society of Obstetrics and Gynecology has no specific information about nutrition during early pregnancy except folic acid.
